# Bacterial, Archaea, and Viral Transcripts (BAVT) Expression in Gynecological Cancers and Correlation with Regulatory Regions of the Genome

**DOI:** 10.3390/cancers13051109

**Published:** 2021-03-05

**Authors:** Jesus Gonzalez-Bosquet, Silvana Pedra-Nobre, Eric J. Devor, Kristina W. Thiel, Michael J. Goodheart, David P. Bender, Kimberly K. Leslie

**Affiliations:** 1Division of Gynecologic Oncology, Department of Obstetrics and Gynecology, University of Iowa Hospitals and Clinics, Iowa City, IA 52242, USA; silvana-pedranobre@uiowa.edu (S.P.-N.); michael-goodheart@uiowa.edu (M.J.G.); david-bender@uiowa.edu (D.P.B.); 2Holden Comprehensive Cancer Center, University of Iowa Hospitals and Clinics, Iowa City, IA 52242, USA; eric-devor@uiowa.edu (E.J.D.); kimberly-leslie@uiowa.edu (K.K.L.); 3Department of Obstetrics and Gynecology, University of Iowa Hospitals and Clinics, Iowa City, IA 52242, USA; kristina-thiel@uiowa.edu

**Keywords:** metagenomics, high grade serous ovarian cancer, endometrioid endometrial cancer, bacterial, archaea and viral transcripts

## Abstract

**Simple Summary:**

Microorganisms are found in all human tissues. Some of them are responsible for cancer formation. In our study we found gene expression from bacteria, archaea, and viruses in the upper female genital tract and this expression was associated with ovarian and endometrial cancer. We also found that the expression from these organisms may be involved in regulatory mechanisms of infection and cancer formation. Some of the processes associated with these organisms may affect cancer heterogeneity and be potential targets for cancer therapy.

**Abstract:**

Bacteria, archaea, and viruses are associated with numerous human cancers. To date, microbiome variations in transcription have not been evaluated relative to upper female genital tract cancer risk. Our aim was to assess differences in bacterial, archaea, and viral transcript (BAVT) expression between different gynecological cancers and normal fallopian tubes. In this case-control study we performed RNA sequencing on 12 normal tubes, 112 serous ovarian cancers (HGSC) and 62 endometrioid endometrial cancers (EEC). We used the *centrifuge* algorithm to classify resultant transcripts into four indexes: bacterial, archaea, viral, and human genomes. We then compared BAVT expression from normal samples, HGSC and EEC. *T*-test was used for univariate comparisons (correcting for multiple comparison) and *lasso* for multivariate modelling. For validation we performed DNA sequencing of normal tubes in comparison to HGSC and EEC BAVTs in the TCGA database. Pathway analyses were carried out to evaluate the function of significant BAVTs. Our results show that BAVT expression levels vary between different gynecological cancers. Finally, we mapped some of these BAVTs to the human genome. Numerous map locations were close to regulatory genes and long non-coding RNAs based on the pathway enrichment analysis. BAVTs may affect gynecological cancer risk and may be part of potential targets for cancer therapy.

## 1. Introduction

An estimated 15–20% of cancers worldwide are linked to viral, parasitic, or bacterial infections and they were responsible for 2.2 million cancer deaths in 2012 [1]. Some of the best examples of cancer-associated infectious agents are hepatitis B virus (HBV), human papillomavirus (HPV), Epstein-Barr virus (EBV), human immunodeficiency virus (HIV), and Helicobacter pylori. A subset of these viruses are known to integrate into the human genome [2]. H. pylori association with carcinogenesis is one of the best understood [3], but there are also other microbes that have been associated with cancer [4], including bladder [5], and colon [6,7,8]. Some of these bacteria associated with cancer may alter the microenvironment to favor tumor formation or favor certain alterations in specific cancers [3].

Several studies, including the Human Microbiome Project Consortium [9,10], have studied the human microbiome (microbial community occupying the human body). These studies included the normal microbiome of the vaginal environment. However, few of them studied the upper genital tract, and even fewer studied gene expression of these organisms (or metagenomics and metatranscriptomics) in the upper tract [11,12].

There is evidence of a microbiome in the female genital tract with indications of gene expression from these microorganisms. However, to date, microbiome variations in transcription have not been evaluated relative to upper genital tract cancer risk. Our hypothesis is that bacteria, archaea, and viruses influence the risk for cancer in the female genital tract, specifically tubal/ovarian and endometrial cancer. This influence also may affect the heterogeneity of these cancers and their distinct responses to therapy. To test this hypothesis, we first assessed differences in bacterial, archaea, and viral transcripts’ (BAVT) expression between normal tube and ovarian cancer. Then compare differences of BAVT expression between different gynecological cancers, ovarian and endometrial. Finally, we assessed the correlation of significant BAVT with gene and long non-coding RNA (lncRNA) expression and various regulatory processes.

## 2. Materials and Methods

This is a retrospective nested case-control study that used clinical and genomic information to create and compare classification of metagenomic sequences from high-grade ovarian and endometrioid endometrial cancers. Due to increasing evidence of the genesis of high-grade serous cancer (HGSC) in the fallopian tube, we used normal fallopian tube samples as a normal control for HGSC [13]. We then compare the results from this classification with the classification scheme of the Cancer Genome Atlas (TCGA) gynecological cancer samples: HGSC and EEC.

### 2.1. Clinical Data

Clinical and pathological data were collected from electronic medical records [14]. Only baseline clinical and pathological characteristics were included.

### 2.2. Biological Data

#### 2.2.1. Samples

Genomic DNA (gDNA) and total cellular RNA were purified from flash frozen tumor (ovarian and endometrial) and normal tissues stored in the Department of Obstetrics and Gynecology Gynecologic Oncology Bank (IRB, ID#200209010) which is part of the Women’s Health Tissue Repository (WHTR, IRB, ID#201809807). A separate approval was given by the University of Iowa (UI) Institutional Review Board (IRB, ID#201202714) to collect 20 normal fallopian tube samples in coordination with the University of Iowa Tissue Procurement Core Facility to be used as controls. Tubal samples came from the junction of the ampullary and fimbriated end of fallopian tubes of volunteers without any family or personal history of cancers who were scheduled to undergo salpingectomy for benign indications (mainly sterilization). No patient indicating a personal or family history of cancer was included. All tissues archived in the WHTR were originally obtained from adult patients under informed consent in accordance with University of Iowa IRB guidelines.

#### 2.2.2. DNA and RNA Purification and Sequencing

The WHTR contains more than 60,000 biological samples, including more than 2500 primary gynecologic tumors [15]. All tissues in the WHTR are collected under informed consent in accordance with the University of Iowa IRB guidelines (IRB Number 200910784 and IRB Number 200209010). Of the 193 patients identified in the original HGSC panel, we were able to obtain 112 tumor tissues with sufficient RNA yield and quality for analysis [16]. Similarly, of the 126 patients identified in the original endometrial endometrioid cancer panel, we were able to obtain 62 primary tumor tissues with sufficient RNA yield and quality for analysis [17]. From the 20 original normal fallopian tube samples, 12 had sufficient RNA yield and quality for analysis [16] (Figure 1). DNA also was extracted from 10 of these normal Fallopian tubes for analysis. Tumor samples were collected, reviewed by a board-certified pathologist, and flash frozen. HGSC diagnosis was confirm in paraffin. Specimens had less than 30% of necrosis.

Total cellular RNA was purified from primary tumor tissue using the mirVana RNA purification kit following manufacturers’ instructions (Thermo Fisher, Waltham, MA, USA). Yield and quality of purified cellular RNA was assessed using a Trinean DropSense 16 spectrophotometer and an Agilent Model 2100 bioanalyzer. Only RNAs with an RNA integrity number (RIN) [18] greater than or equal to 7.0 were selected for RNA sequencing. Equal mass total RNA (500 ng) was quantified by Qubit measurement (Thermo Fisher). Each qualifying tumor was fragmented, converted to cDNA and ligated to bar-coded sequencing adaptors using Illumina TriSeq stranded total RNA library preparation (Illumina, San Diego, CA, USA).

Genomic DNAs were purified from frozen tumor tissues using the DNeasy Blood and Tissue Kit according to manufacturer’s (QIAGEN GmbH, Hilden, Germany) recommendations.

Molar concentrations of the indexed libraries were confirmed on the Agilent Model 2100 bioanalyzer and libraries were then combined into equimolar pools for sequencing. The concentration of the pools was confirmed using the Illumina Library Quantification Kit (KAPA Biosystems, Wilmington, MA, USA). Sequencing for both RNA and DNA was then carried out on the Illumina HiSeq 4000 genome sequencing platform using 150 bp paired-end SBS chemistry. All library preparation and sequencing were performed in the Genome Facility of the University of Iowa Institute of Human Genetics (IIHG). Quality control (QC) of RNA sequencing experiments (RNA-seq) were performed to minimalize technical biases.

#### 2.2.3. Metagenomics Classification

Classification is different from alignment in that classification is performed on a large set of genomes as opposed to just one reference genome when alignment is done [19]. Before the actual classification is done, an index with the set of genomes to use for the classification must be built. We built an index with the genomes of bacterial, archaea, viral, and human species. *Centrifuge* is a metagenomics classifier that can store and manage large number of genomes, taxonomical mappings, and tress [19].

We obtained *fastq* files from RNA-seq experiments of all samples. Then, we applied the *Centrifuge* algorithm to extract reads from *fastq* files and classify resultant transcripts in four possible indexes: bacterial, archaea, viral and human genomes. After the classification, we took the transcripts counts for all indexes and used DESeq2 package to import, normalize, and log2-transform the data for analysis [20]. There were 9957 different transcripts resulting from the classification, including different Taxonomic ranks. For the study we compare only transcripts at the *species* Taxonomic rank: a total of 5042 unique transcripts for HGSC, EEC and control (fallopian tubes) samples.

### 2.3. Statistical Analysis

Comparisons between the number of BAVTs found in normal fallopian tube and HGSC samples were assessed with chi-square. A *p*-value < 0.05 was considered significant.

#### 2.3.1. Association of BAVT with HGSC 

Comparison of BAVT between normal fallopian tubes and HGSC samples was performed with multiple t-test of normalized, log2-transformed transcript counts. To account for multiple comparisons, differences of transcripts (BAVT) expressions between the classes (HGSC and normal tube) at the univariate significance level of *p* < 0.005 were considered significant [21]. Then, we performed a multivariate *lasso* regression analysis to assess the independent BAVTs associated with cancer was built introducing significant variables in the univariate analysis. *Lasso* is a multivariate regression method that allows simultaneous selection and estimation of the effects of variables, while accounting and adjusting for confounding factors [22].

#### 2.3.2. Differences of BAVT Expression between HGSC and EEC

Comparison between BAVT among HGSC and EEC samples were performed as previously (*t*-test). Similarly, multiple comparisons were accounted for.

#### 2.3.3. Validation of Differential BAVT between HGSC and EEC in TCGA Dataset

Validation of BAVT expression was performed using the TCGA HGSC and EEC databases. BAM files were downloaded from TCGA website. Then, these files were transformed to *fastq* format with *bedtools* a suite of tools for genomic analysis [23]. Then fastq files were processed as previously to obtain normalized, log2-transformed BAVTs from HGSC and EEC samples form TCGA. For validation of previous results, those BAVTs that were considered significant in previous comparisons were assessed in TCGA BAVT expressions.

#### 2.3.4. Correlations between BAVT Expression and Gene and lncRNA Expressions

Correlations between BVAT and gene and lncRNA expression were performed using Spearman’s rank correlation test, as the expression between these genomic elements is not completely independent from one another. Statistical significance was assessed with p-value and Bonferroni correction for multiple comparisons [21].

#### 2.3.5. Power Calculation

If we take the 50th percentile of the variance distribution, we will need 13 samples per group, for the analysis to have 80% power to find transcripts with a mean difference of 1 in expression (log2-transformed) between classes and 0.005 Type 1 error. For the 75th percentile of the variance distribution we would need 23 samples per group.

### 2.4. Bioinformatics

#### 2.4.1. Mapping Significant Transcripts to the Human Genome

To identify exactly where significant BAVT is mapped in the human genome, we first downloaded reference sequences (from the NCBI, or National Center for Biotechnology Information) of those species that were detected and found to be significant in previous analyses. These sequences were aligned with *fastq* files from ovarian cancer patients using another aligning method, HISAT2 software [24]. Concordant aligned pairs were selected using the SAMtools software [25]. A concordant pair aligns with the expected relative mate orientation and with the expected range of distances between mates. Then, selected alignments with BAVT were blasted against the human genome (hg38 version) with *BLASTN*, megablast option [26]. The best alignments were mapped with UCSC Genome Browser into the human genome (hg38) at the chromosomal level. Genes and transcripts were included in the same UCSC visualizing window that the mapped sequences were considered to be in the vicinity (around 25kb per side).

To assess expression count, RNA-seq reads were mapped and aligned to the human reference genome (version hg38) using STAR, a paired-end enabled algorithm [27]. BAM files were produced after alignment and featureCount to measure gene expression from BAM files [28]. LncRNA were also determined using BAM files as input and processing the files with UClncR [29]

#### 2.4.2. Pathway Enrichment Analysis

To identify over-represented and significant pathways among the selected list of genes we used clusterProfiler (R project), an integrated and curated “knowledge-based” platform that uses KEGG databases [30,31]. The p-value of significant associated pathways represents the probability that a particular gene of an experiment is placed into a pathway by chance, considering the numbers of genes in the experiment, and total genes across all pathways.

Except for *Centrifuge*, most analyses were performed using R statistical package for statistical computing and graphics (www.r-project.org, accessed on 20 January 2020) as background, using Bioconductor packages as open-source software for bioinformatics (bioconductor.org, accessed on 20 January 2020).

## 3. Results

### 3.1. Association of BAVT with HGSC

We found 4.7% BAVT in normal tubal samples and 3.1% in HGSC, *p* < 0.001 (Figure 2A, Appendix A). There were 202 BAVT that were significant in the univariate *t*-test analysis. These differences are observed with a heatmap (Figure 2B). Species with highest difference are detailed in Figure 2C.

We introduced all significant differentially expressed BAVT between normal tubes and HGSC into a multivariate *lasso* regression model. The model resulted in 12 species independently associated with HGSC (Figure 3A,B). As previously, the majority of BAVT were more highly expressed in the normal tissue than in cancer samples (Figure 3C).

### 3.2. Differences of BAVT Expression between HGSC and EEC

Then we compared BAVT expression levels between HGSC and EEC and found 93 BAVTs differentially expressed between both types of gynecologic cancers. In Figure 4 we present a heatmap of the differentially expressed BAVTs by location (Figure 4A). When we compared the expression of the twelve BAVT species independently expressed between HGSC and normal tubes between HGSC and EEC samples, 7 of them were also significant: Pusillimonas sp. ye3, Riemerella anatipestifer, Salinibacter ruber, Bacillus tropicus, Nostocales cyanobacterium HT-58-2, Orgyia pseudotsugata, and Corynebacterium pseudotuberculosis. We represented BAVT expression of these 12 independently expressed BAVT between HGSC and normal tubes in all samples to appreciate expression patterns (Figure 4B). Furthermore, when the normalized mean expression of independently significant BAVT were plotted by sample anatomical location, normal fallopian tube had the highest overall expression. Between EEC and HGSOC samples, the majority of BAVT expression was higher in EEC samples and expression in HGSC samples was the lowest (Figure 4C).

To evaluate whether genomic material from these microorganisms was present only in the transcriptome or if it was also present at the genetic (DNA) level, we extracted DNA also from some of the normal fallopian tubes (*n* = 10) and performed whole genome sequencing. To classify tubal germline DNA in bacterial, archaea, or viral (BAV) we applied the *centrifuge* classification algorithm. We then assessed the presence of BAV germline DNA copies with BAVT expression in normal tubes and we found over 10,660 different BAV genomic sequences (Figure 5). We looked for those DNA sequences from all the BAVT that were significant in the univariate analysis comparing tubes and HGSC, *N* = 202 (Figure 5A). There were only 10 BAVT significant in the univariate analysis that had no genomic DNA copies (Figure 5B), but all BAVT significant in the multivariate analysis had their respective DNA copy (Figure 5C).

### 3.3. Validation of BAVT Analysis in TCGA Dataset

To assess whether our findings are also observed in other genomic databases, we downloaded TCGA datasets for EEC and HGSC, converted BAM files into fastq format and applied the *centrifuge* classification algorithm. We observed that 0.8% of all transcripts in the combined TCGA dataset (EEC = 407 and HGSC = 373) were BAVT (Figure 6A). We also identified 91 out of the 93 significant BAVT in the comparison between EEC and HGS performed in samples from the UI: 88 were significant, with a *p*-value < 0.05 (Figure 6B) with an accuracy of 64% (95% CI: 59%, 71%, Figure 6C). All BAVT significant in the multivariate analysis between tube and HGSC were also significant in TCGA analysis of EEC versus HGSC (all with *p* < 0.001, Figure 6D). We note that RNA-seq of TCGA samples was done on 75 mers while for the UI RNA-seq we used 150 mers. This may have affected the background noise of the measurement and, potentially, the accuracy.

### 3.4. Mapping Significant BAVT and Correlation with Gene and lncRNA Expression

To investigate the potential origins of these BAVT, we mapped these sequences to the human genome, hg38 version. First, we mapped independently significant BAVT from the HGSC vs. tubal multivariate analysis, on reference sequences downloaded from the NCBI (Table 1). The best concordant aligned sequences were blasted against the human genome and then mapped into the UCSC human genome (Appendix A and Appendix A). Known genes and transcripts in the vicinity of the mapped sequences were identified: those that were located within the margins of the blasted sequence, and those that were close to that location. See Appendix A for more information about the proximity of these loci, and Appendix A for detailed mapping of these BAVTs on hg38 chromosomes.

The majority of these BAVT were located within or close to genes or lncRNAs. Seven BAVTs, Bacillus megaterium, Cutibacterium acne, Orgyia pseudotsugata, Mycobacterium shigaense, Bacillus tropicus, Pusillimonas sp., were mapped close to lncRNAs: AL591845.1, WARS2-AS1, LINC01594, AC091685.1, AF279873.3, LINC00350, LINC01224, FP671120.4, FP236383.3, AL807742.1, AC114814.3, AC092691.3, AC137810.1, LINC01170, LINC02553, AC090680.1, LINC00399, AL163541.1, LINC00273. Other BAVTs were mapped within or in close proximity to 38 independent genes (Appendix A). Several BAVTs mapped in multiple loci in the genome. To evaluate the association between BAVT mapping and gene/lncRNA expression, we correlated the expression of mapped BAVT with the expression of lncRNA close by and with expression of those 38 genes, as explained in the Methods section (Figure 7). Then, we evaluated the association between significantly correlated lncRNA in Figure 7A (LINC02553, AL591845.1, and AL163541.1) with all expressed genes (Table 2).

### 3.5. Pathway Enrichment Analysis

All significantly correlated genes with either mapped BAVT or with significant lncRNA (LINC02553, AL591845.1, AL163541.1) were introduced in an enrichment pathway analysis. Overrepresented and statistically significant pathways in this list of genes are displayed in Table 3. Most of these pathways are involved in infectious processes, including the novel Coronavirus disease—COVID-19 pathway (representation of KEGG pathways are in Appendix A).

## 4. Discussion

Bacterial, archaea, and viral transcripts’ (BAVT) expression is found in RNA-seq experiments of samples from normal tissues and cancers [11,32,33,34]. Most of the metagenomic and metatranscriptomics studies are focused on cancers from the gastrointestinal tract [35], or skin [36], or head/neck regions [37]. However, the presence of microorganisms and BAVT in the female upper genital tract has recently been reported [11,12,34]. These studies not only have demonstrated the existence of microorganism in the female genital tract, but also have observed that components of this ecosystem gradually change from the vagina to the peritoneal fluid [12]. Also, there were differences in the composition of transcripts found in these areas which showed genes involved in metabolism, replication and repair, membrane transport, and drug resistance in upper areas, whereas in the lower areas predominated genes were involved in translation, energy metabolism, and cofactors and vitamins metabolism. Though vaginal specimens were well represented, there were few samples from the upper tract and that limited the scope and generalizability of their conclusions [12]. With the present study we add further evidence of the presence of BAVT in the upper genital tract in normal and cancer samples from patients. Further, we found genetic copies of bacterial, archaea, and viral DNA in benign fallopian tube samples. It must be noted that in our study all samples were obtained during surgery, so they were collected under clean conditions. We do not completely understand the origin of these BAVT, but we have determined that there are loci in the human genome that seem to preferentially harbor genetic material from these microorganisms. We do not know as yet if they are functional or if they are transcribed. BAVTs could potentially originate partially from normal genital tract flora and partially from inserted microbial genetic material. Some of the BAVT that were significant in the univariate analysis had no reciprocal DNA copy in the genome of the tube, supporting the idea of external BAVT production.

Some of the described microorganisms associated with HGSC and EEC have been associated previously with cell proliferation, some types of cancer or even cancer treatment. The Orgyia pseudotsugata nuclear polyhedrosis virus expression of proteins that inhibit apoptosis in immature thymocytes have also been implicated in cell division, cell cycle regulation, and cancer [38]. Cutibacterium acnes (formerly Propionibacterium acnes) have also been linked to the development of prostate cancer [39]. Also, certain types of splenic lymphomas have been associated to disseminates infections of Mycobacterium shigaense [40]. Lastly, a lipopeptide from the bacteria Bacillus megaterium seems to be a potent chemosensitizer in docetaxel resistant breast cancer cells by reducing the AKT signaling pathway [41].

We have observed that the amount of BAVT expression is associated with the presence of gynecological cancer and may also be important to the location of the cancer. BAVT expressions in the normal tissue were the highest and were decreased in EEC and even more in HGSC samples. Moreover, potentially, BAVTs could originate from the host genome and from normal flora within the genital tract. The balance between both sources could determine the quantity of transcript present, and that could turn out to be vital in modulating the risk of cancer. Any condition that alters this balance may also influence specific cancer risks or may modulate a variety of genomic features that may affect tumor heterogeneity. For example, in epidemiological studies it has been observed that tubal ligation, that usually blocks tubal lumen and passage, decreases the risk for ovarian cancer [42,43]. Also, endometriosis has been postulated as a risk factor for ovarian cancer [44,45]. Inflammation due to tubal endometriosis is a known factor involved in ovarian cancer carcinogenesis, mainly through DNA mutation mediated by free radicals associated with inflammatory process [46]. Further alteration of the microenvironment through microbiome and BAVT dysregulation may be synergistic or inhibitory of these mechanisms. Another example is the use of powdered talc in the genital area. This chemical may alter the microenvironment subsequently elevating the risk of cancer [47]. Finally, in observational studies patients with ovarian cancer or predisposition for ovarian cancer (germline *BRCA1* mutation) were associated with a specific microbiome in the vagina/cervix [48]. Further studies are needed to identify the exact origin of these BAVT (host or microorganisms), the balance that is needed for organ homeostasis, and the unbalance that may lead to alterations.

Significant BAVT in multivariate analysis comparing normal tube versus HGSC were correlated with gene expression and expression of some regulatory lncRNAs. In the pathway analysis, these associated genes seem to play a role in pathways of infection and intracellular signaling which also are important in cancer genesis [49]. *IFNA21* is one of the genes correlated with BAVT and is a member of the alpha interferon gene cluster that are produced in response to viral infection. Similarly, *IFNA21* is an important component of the PI3K-AKT/mTOR pathway and was significantly correlated with the lncRNA AL163541.1, which in turn is significantly associated with the presence of Orgyia pseudotsugata transcripts. Also, *IFNA21,* as part of the alpha interferon gene cluster, mediates the immune response and interfere with viral replication. It has been observed that deletions of chromosome 9 short arm, which harbors a cluster of these interferon genes, may occur in acute lymphoblastic leukemia and in gliomas. Deletions are observed in lower frequency in lymphomas, melanomas, lung cancers, and other solid tumors [50]. *ITGA3* has been implicated in the infectivity and associated signaling pathways in Kaposi’s sarcoma-associated herpesvirus HHV-8 entry into target cells [51]. *JAK1* is one of the components of the PI3K-Akt signaling and JAK/STAT3 signaling pathways. The PI3K-AKT/mTOR pathway has a critical malignant transformation of human tumors [52] and, specifically, is one of the major pathways that is aberrantly activated in ovarian cancer and associated with tumor progression and poor prognosis in patients with ovarian cancer [53]. This pathway is a potential target for new therapeutic agents [54]. PI3K-AKT/mTOR pathway is downregulated by *PTEN*, a tumor suppressor. Mutations in *PTEN* are the most frequent genetic alteration in endometrial cancer [55]. In cell cultures, *PTEN* mutated cells seem to be more sensitive to poly ADP-ribose polymerase (PARP) inhibitors [55]. Some viral infections, like SARS-CoV2, also seem to hyperactivate the JAK1/2-STAT1 signaling system, especially in critical patients [56]. The hepatocyte growth factor (HGF) has been known to play a predominant role in many types of human cancers, promoting invasiveness and metastasis of ovarian cancer cultured cells. Interestingly, these effects may be mediated by PARP-1 [57]. Finally, EGFR tyrosine kinase inhibitors (TKI) modulate the antitumoral activity of immune cells. However, long-term exposure of ovarian cancer cells to TKIs may reduce the responsiveness to treatment [58]. Anti-EGFR TKIs may help modulate that resistance with promising results in colon cancer [59]. Pathway analyses could inform how genes correlated with BAVT are associated with signaling pathways. However, we need functional analysis to better understand how BAVTs influence signaling pathways, like JAK/STAT and PI3K-AKT/mTOR. Even more importantly, functional analysis could better inform whether changes in BAVT expression could help in the selection of targeted therapies, like: JAK inhibitors to target the JAK1/STAT3 signaling pathway [53], or mTOR inhibitors [54], PARP inhibitors to target HGF-mediated activation in ovarian cancer [57] or PTEN-mutated cells in endometrial cancer [55], and double TKIs therapy to counteract EGFR TKI resistance in endometrial cancer [59].

A limitation of this study is the retrospective nature of the design. These studies were not originally designed to be a functional analysis of BAVTs expression. Thus, despite statistical significance in the association between BAVTs and cancer and correlation with gene regulation, these results will have to be examined and validated mechanistically. Likewise, it is difficult to determine the exact origin of BAVT expression whether intrinsic (from the host) or extrinsic (by colonizing microorganisms). To better identify and quantify origins, functional analyses are needed that are beyond the scope of this study. A major strength of this study is that the data were collected at a single tertiary medical center, which ensured protocol consistency in sample collection and analysis procedures. In addition, due to the diversity of patients treated at a large tertiary medical center, the samples included in this study likely represent a broad array of the clinical phenotypes of EEC and HGSC. Despite the limited number of normal fallopian tube control samples, tubal RNA-seq analysis findings were validated by determinations of BAVT gene copies with DNA-seq of the same tubal samples. Validation of BAVTs expression comparison between EEC and HGSC was performed in the independent, well-known, and validated TCGA database. Although samples were collected in a clean environment (OR setting), contamination has been reported in laboratory instruments such as spin columns [60]. We do not believe that the results were biased due to contamination. Validation of RNA-seq results with DNA-seq determinations and with validation in an independent dataset along with independent lab processing minimizes the possibility of bias by contamination.

## 5. Conclusions

In summary, we have identified bacterial, archaea, and viral transcripts’ (BAVT) expression in samples from normal fallopian tubes and gynecologic cancers. We find that BAVT expression levels vary between different gynecological cancers. Although confirmation is needed, some of this BAVT expression may originate from genomic material embedded in the human genome. Genes and lncRNAs correlated with BAVTs were associated with infectious and cancer signaling pathways. Some of these pathways associated with BAVT could inform of potential candidates for targeted therapy in future mechanistic analysis.

## Figures and Tables

**Figure 1 cancers-13-01109-f001:**
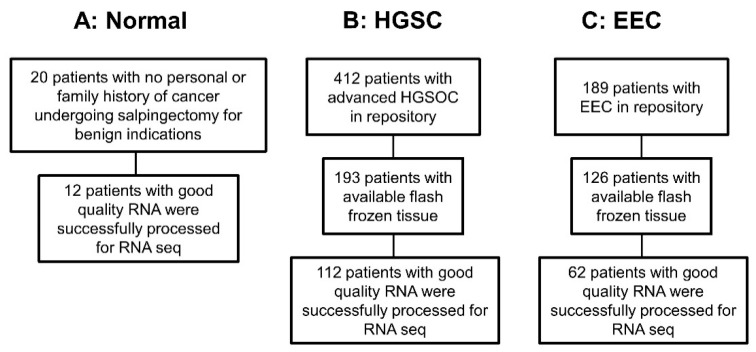
Flowchart of patients included in the analysis: (**A**) samples from patients with normal fallopian tubes, and no risk factors for cancer. (**B**) Samples from patients with high grade serous ovarian cancer (HGSC). (**C**) Samples from patients with endometrioid endometrial cancer (EEC).

**Figure 2 cancers-13-01109-f002:**
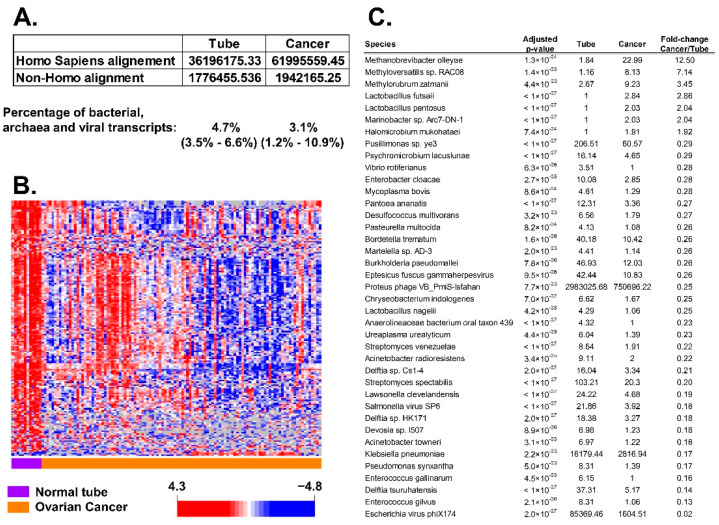
Differential transcript expression of bacterial, viral, and archaea (BAVT) organisms between normal tube tissue and HGSC: (**A**) average of aligned transcripts in normal tube and HGSC: percentage of non-human (BAVT) among the total aligned RNA (chis square *p*-value < 0.001). (**B**) Heatmap plot of significant differentially expressed BAVT after filtering (>50% missing values). Univariate analysis with multiple comparison adjustment, level of significance is *p* < 0.005: 202 BAVT were different. In the heatmap there is a general decrease of non-human transcripts in ovarian cancer in comparison with expression in normal tubes. Orange samples: HGSC; purple samples: normal tube. (**C**) Significant BAVT with largest differential expression between tube and cancer.

**Figure 3 cancers-13-01109-f003:**
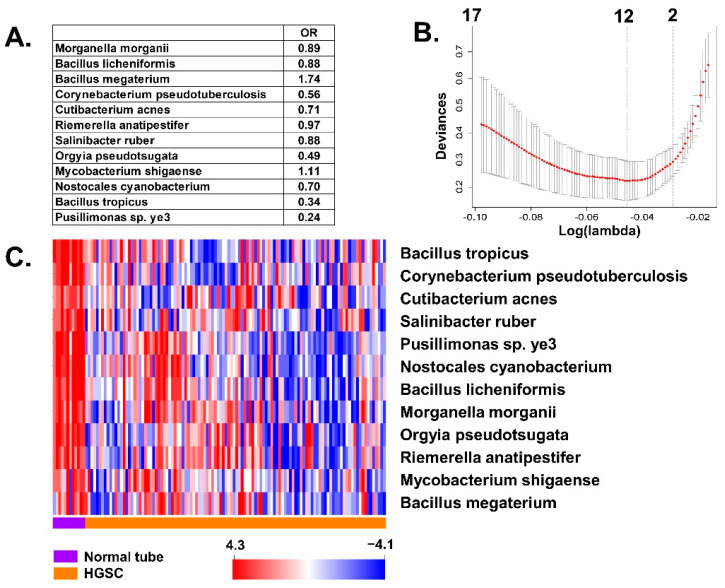
Multivariate lasso regression analysis of BAVT expression and association with cancer: (**A**) Best model with significant transcripts and their effect: decreases (<1) or increases (>1) in these species’ BAVT are independently associated with cancer. (**B**) Best model with 12 variables (top of the panel): The plot displays the cross-validation error according to the log of lambda. The left dashed vertical line indicates that the log of the optimal value of lambda is approximately −0.05, which is the one that minimizes the prediction error. This lambda value will give the most accurate model. (**C**) Heatmap plot of independently significant BAVT: orange: ovarian cancer; purple: normal tube.

**Figure 4 cancers-13-01109-f004:**
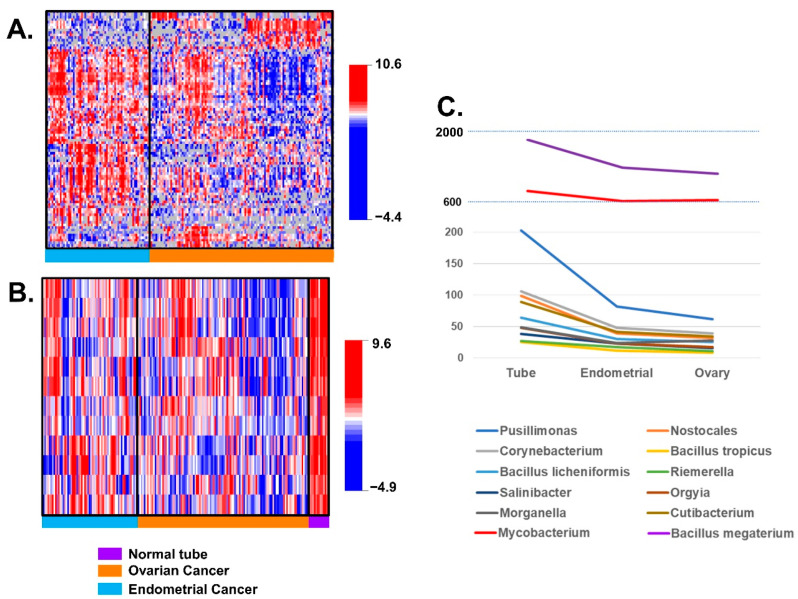
Differential BVAT expression between HGSC and EEC samples: (**A**) heatmap plot of significant differential level of expression of BAVT (*p* < 0.005, for multiple comparison adjustment): blue samples: endometrial cancer; orange samples: ovarian cancer. In general, endometrial cancer level of expression of non-human transcripts was higher than ovarian cancer. 93 of these BAVT were significantly different. (**B**) Heatmap plot of independently significant non-human transcripts: EEC BAVT expressions are between HGS and normal tubes (blue samples: endometrial cancer; orange samples: ovarian cancer; purple samples: normal tube). (**C**) Normalized mean expression of independently significant BAVT represented by anatomic location: fallopian tube, endometrium and ovarian. Tube has the highest expression, followed by the EEC except for 2 transcripts, Morganella morganii and Mycobacterium shigaense, where ovarian expression is slighter lower than endometrium.

**Figure 5 cancers-13-01109-f005:**
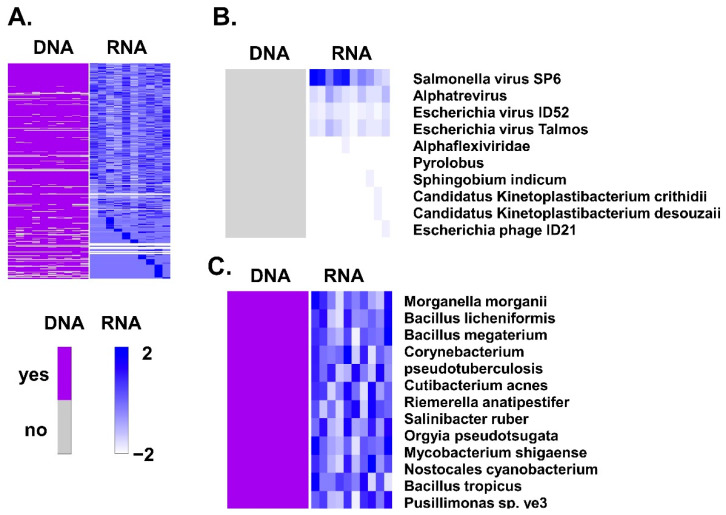
Correlation between bacterial, archaea, or viral (BAV) DNA present in the normal (control) tube and their respective BAVTs: (**A**) BVA sequences in DNA of normal fallopian tubes are represented in left side of the panel. Only present or not present was assessed. We determined the number of copies transcribed from this DNA in the same 10 samples (right side of the panel). Only BAVT (and respective BAV DNA) that were significant in the univariate analysis are represented. (**B**) There were 10 BAVT transcripts significant in the univariate analysis that had no genomic DNA copy (grey color) in the normal tube. (**C**) All independently significant BAVT in the multivariate analysis had genomic copies in the normal tubal DNA.

**Figure 6 cancers-13-01109-f006:**
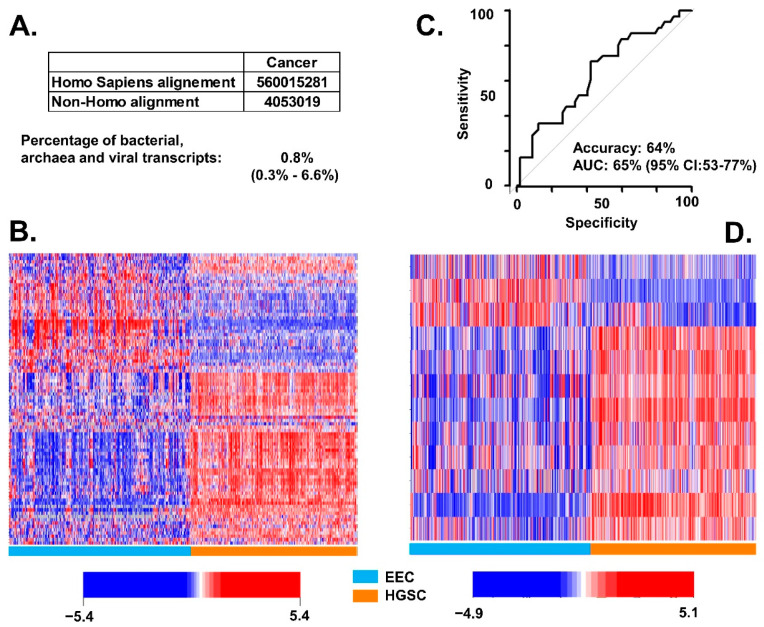
Differential BAVT expression between HGSC and EEC cancer in TCGA. (**A**) Average of aligned transcripts in all TCGA dataset: percentage of non-human (BAVT) among the total aligned RNA. (**B**) Heatmap plot of BAVT transcripts in TCGA patients that were significant in the UI univariate analysis (HGSV vs. EEC, *N* = 88). Cyan samples: EEC; orange samples: HGSC. In the heatmap: blue represents less copies of transcripts; red represents more transcripts. (**C**) Calculated accuracy and AUC of 91 BAVT present (out of 93 in UI) in the comparison between EEC and HGS in TCGA dataset. (**D**) Heatmap plot of BAVT transcripts in TCGA patients that were significant in the UI multivariate analysis of tube vs. HGSC.

**Figure 7 cancers-13-01109-f007:**
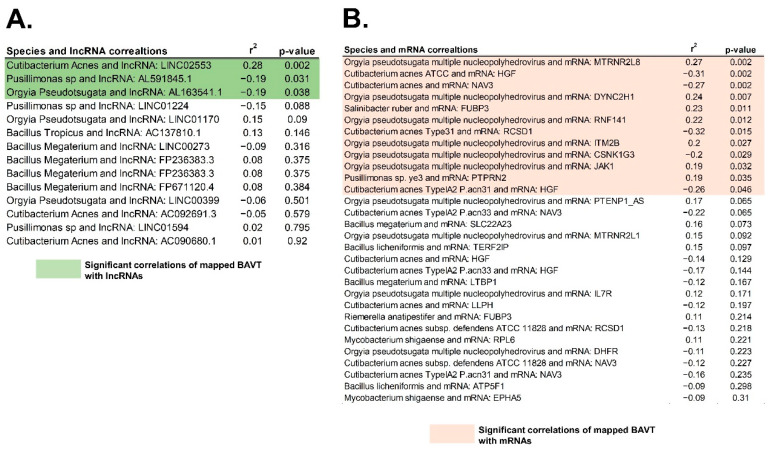
Correlation of mapped and significant BAVT with lncRNA and gene expression: (**A**) Correlation of mapped BAVT and lncRNA: highlighted in green are the significant correlations (*p* < 0.05). (**B**) Correlation between mapped BAVT and 37 close-by expressed genes (mRNA): in orange, significant correlations (*p* < 0.05).

**Table 1 cancers-13-01109-t001:** Significant species aligned to normal tube and HGSC samples. Represented are the species nucleotides that better aligned with the RNA-seq transcripts of the HSGC samples. The last two columns contain averages of aligned bases. TaxID: Taxonomy species ID; Nucleotides: NCBI accession numbers of the reference sequence of significant species.

Species	TaxID	Nucleotides	Tube	Cancer
Morganella morganii	582	NZ_CP033056.1	107.72	105.56
Bacillus licheniformis	1402	NZ_CP021669.1	75.95	76.12
Bacillus megaterium	1404	NZ_CP026740.1	121.36	122.05
		NZ_CP026741.1	133.93	132.99
Corynebacterium pseudotuberculosis	1719	NZ_CP046731.1	199.59	201.63
		NZ_CP046732.1	199.59	201.63
Cutibacterium acnes	1747	NZ_AP019664.1	262.37	253.21
		NZ_CP012351.1	265.00	258.35
		NZ_CP012352.1	301.50	266.20
		NZ_CP012354.1	266.00	243.00
		NZ_CP012355.1	261.00	265.87
		NZ_CP012647.1	264.69	259.79
Riemerella anatipestifer	34,085	NZ_CP029760.1	125.79	141.15
		NZ_CP045564.1	84.00	89.71
		NZ_LT906475.1	85.11	79.47
Salinibacter ruber	146,919	NZ_CP030356.1	23.20	28.50
		NZ_CP030716.1	26.94	27.13
Orgyia pseudotsugata multiple nucleopolyhedrovirus	262,177	NC_001875.2	148.10	160.98
Mycobacterium shigaense	722,731	NZ_CP022927.1	127.22	126.70
Nostocales cyanobacterium HT-58-2	1,940,762	NZ_CP019636.1	58.18	57.39
Bacillus tropicus	2,026,188	NZ_CP041081.1	223.61	212.87
Pusillimonas sp. ye3	2,028,345	NZ_CP022987.1	89.07	87.06

**Table 2 cancers-13-01109-t002:** Correlation of significant lncRNAs with all gene expression. LncRNAs that were significantly correlated with mapped BAVT: LINC02553 with Cutibacterium Acnes, AL591845.1 with Pusillimonas sp., and AL163541.1 with Orgyia Pseudotsugata, were latter correlated with gene expression to understand the possible association between both expressions.

lncRNA	mRNA	r^2^	*p*-Value
AL163541.1	MIR4539	0.32	0.0002
AL163541.1	FXYD7	0.32	0.0003
AL163541.1	ARL9	0.31	0.0004
AL163541.1	KCNQ5-AS1	0.31	0.0004
AL163541.1	ZNF433	−0.30	0.0008
AL163541.1	MIR933	0.29	0.0010
AL163541.1	FXYD5	0.29	0.0010
AL163541.1	OR8B8	0.29	0.0012
AL163541.1	MYCNOS	−0.29	0.0013
AL163541.1	RYBP	−0.29	0.0013
AL163541.1	RNU6-69P	−0.28	0.0014
AL163541.1	SLC22A6	−0.28	0.0016
AL163541.1	ANO10	−0.28	0.0016
AL163541.1	PANO1	0.28	0.0016
AL163541.1	LOC642366	0.28	0.0019
AL163541.1	POLM	0.27	0.0020
AL163541.1	SMC6	−0.27	0.0029
AL163541.1	MIR4733	−0.26	0.0030
AL163541.1	ZNF90	−0.26	0.0033
AL163541.1	TTC32	−0.26	0.0033
AL163541.1	VTRNA1-1	−0.26	0.0034
AL163541.1	MIR466	0.26	0.0035
AL163541.1	IFNA21	0.26	0.0036
AL163541.1	DKKL1	−0.26	0.0036
AL163541.1	MYCN	−0.26	0.0038
AL163541.1	MIR5582	−0.26	0.0039
AL163541.1	TAF1B	−0.26	0.0039
AL163541.1	RDX	−0.26	0.0039
AL163541.1	RPS12	0.26	0.0041
AL163541.1	KRTAP24-1	0.26	0.0041
AL163541.1	PDCD6IPP2	−0.26	0.0041
AL163541.1	GJC2	0.25	0.0043
AL163541.1	ITGA3	0.25	0.0044
AL163541.1	ANXA8	0.25	0.0044
AL163541.1	MIR3944	0.25	0.0045
AL163541.1	MIR3126	0.25	0.0047
AL163541.1	ZNF676	−0.25	0.0048
AL163541.1	FAM50B	0.25	0.0049
AL163541.1	MESDC2	−0.25	0.0050

**Table 3 cancers-13-01109-t003:** Pathway enrichment analysis with *clusterProfiler* (R project) with all significant correlated genes between mapped BAVT and/or significant lncRNAs (LINC02553, AL591845. AL163541.1). Based on KEGG Database information.

ID	Description	*p*-Value	Symbols
hsa05168	Herpes simplex virus 1 infection	0.002	ZNF433/ZNF90/IFNA21/ZNF676/JAK1
hsa04151	PI3K-Akt signaling pathway	0.003	IFNA21/ITGA3/HGF/JAK1
hsa05171	Coronavirus disease—COVID-19	0.008	IFNA21/RPS12/JAK1
hsa01521	EGFR tyrosine kinase inhibitor resistance	0.009	HGF/JAK1
hsa05165	Human papillomavirus infection	0.021	IFNA21/ITGA3/JAK1
hsa03450	Non-homologous end-joining	0.024	POLM
hsa05162	Measles	0.027	IFNA21/JAK1
hsa05160	Hepatitis C	0.033	IFNA21/JAK1
hsa04217	Necroptosis	0.034	IFNA21/JAK1
hsa04630	JAK-STAT signaling pathway	0.035	IFNA21/JAK1
hsa05161	Hepatitis B	0.035	IFNA21/JAK1
hsa05164	Influenza A	0.039	IFNA21/JAK1
hsa05152	Tuberculosis	0.043	IFNA21/JAK1
hsa04621	NOD-like receptor signaling pathway	0.043	IFNA21/JAK1
hsa05167	Kaposi sarcoma-associated herpesvirus infection	0.049	IFNA21/JAK1

## Data Availability

Clinical data is not publicly available due to patient privacy. Datasets with RNA-seq can be browsed by their accession number: GSE156699. The validation part of this study was performed in silico, with de-identified publicly available data. All data from TCGA is available at their website: https://portal.gdc.cancer.gov/, accessed date 20 January 2020. Software utilized by this study is also publicly available at Bioconductor website: http://bioconductor.org/, accessed on 20 January 2020.

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
