# Peer review of "Bacterial, Archaea, and Viral Transcripts (BAVT) Expression in Gynecological Cancers and Correlation with Regulatory Regions of the Genome"

_cancers, 2021, doi:10.3390/cancers13051109_

Round 1

Reviewer 1 Report

The manuscript (cancers-1084595) entitled “Bacterial, archaea and viral transcripts (BAVT) expression in gynecological cancers and correlation with regulatory regions of the genome” by Bosquet et al is an original research article aimed to assess differences in bacterial, archaea and viral transcripts’ (BAVT) expression between different gynecological cancers and normal fallopian tubes.  

In my opinion the manuscript has several flaws and cannot be accepted in the present version.

Major comments

  • The Introduction is too long and confused. For example, the text from lines 53 to lines 61 is in part repetitive and very similar to the text immediately below and overall it seems to be more indicated for the discussion rather than introduction. I would move it to the Discussion.
  • Moreover, some sentences in the introduction are not clear for me. For example, lines 46-47.
  • Methods: I found not methodologically correct to compare EEC with normal fallopian tube; why the authors did not compare EEC with normal endometrium or benign endometrial hyperplasia? It would be more correct.
  • Methods: Also, the decision to use normal fallopian tube as comparator for ovarian cancer could be debatable. In fact, not all ovarian cancers develop from the fallopian tube (this is a recent theory valid for some but not all cases) but they can originate also from ovarian parenchymal tissue.
  • Moreover, the normal tube is physiologically more involved than other tissues (i.e., ovary and endometrium) in infectious processes; then it seems physiological to finding a higher expression of BAVTs in normal fallopian tubes.
  • Discussion: lines 356-358: how the authors can explain and support these two sentences with a likely opposite meaning.
  • Lines 364-367: inflammation that characterizes the ovarian cancer microenvironment is a well-known factor involved in ovarian cancer carcinogenesis, independently from the changes in BAVTs. Indeed, inflammation is involved in ovarian cancer carcinogenesis mainly through the induction of DNA mutation mediated by the action of free radicals associated to inflammatory process. These mechanisms are well reviewed in Macciò et al Cytokine 2012.
  • Lines 376-378: these sentence that introduce the more detailed list of main pathways below is not completely clear. In fact, although some pathways cited could be involved in an increased cancer risk if they are suppressed (i.e. IFN), others such as JAK and STAT-3 are involved in carcinogenesis and cancer progression and prognosis when they are hyperexpressed (similarly also EGFR and HGF). Therefore, it is not clear how lower expression of these BAVTs in cancers in comparison to normal tubes can be involved in carcinogenesis.
  • Some limitations, notably the risk of microbial contamination of samples, are very concerning and should be avoided by methodologically adequate procedures.
  • Overall, the manuscript is inadequate in methodology and confused both in the Introduction and in the Discussion sections.

Author Response

REVIEWER 1

The manuscript (cancers-1084595) entitled “Bacterial, archaea and viral transcripts (BAVT) expression in gynecological cancers and correlation with regulatory regions of the genome” by Bosquet et al is an original research article aimed to assess differences in bacterial, archaea and viral transcripts’ (BAVT) expression between different gynecological cancers and normal fallopian tubes.  

In my opinion the manuscript has several flaws and cannot be accepted in the present version.

Major comments

  • The Introduction is too long and confused. For example, the text from lines 53 to lines 61 is in part repetitive and very similar to the text immediately below and overall it seems to be more indicated for the discussion rather than introduction. I would move it to the Discussion.

Done

  • Moreover, some sentences in the introduction are not clear for me. For example, lines 46-47.

We have clarified the meaning of the sentence and paragraph.  

  • Methods: I found not methodologically correct to compare EEC with normal fallopian tube; why the authors did not compare EEC with normal endometrium or benign endometrial hyperplasia? It would be more correct.

We intended to compare ovarian cancer and endometrial cancer. We used fallopian tube as control for ovarian cancer. For the present study, we had no availability of normal endometrium for RNA sequencing. Ideally, we would like to have a sample extracted and processed with the same conditions that the cancer samples to be comparable. We are asking for more funding to increase the number of cases and controls for both cancers.

We have clarified this concept throughout the manuscript and also made changes to represent the comparison between HGSC and ECC exclusively. Changes: Introduction, in the objectives description; Methods, section ‘Differences of BAVT expression between HGSC and EEC’; Results, sections ‘3.2. Differences of BAVT expression between HGSC and EEC’ and ‘3.3. Validation of BAVT analysis in TCGA dataset’; and Discussion.

  • Methods: Also, the decision to use normal fallopian tube as comparator for ovarian cancer could be debatable. In fact, not all ovarian cancers develop from the fallopian tube (this is a recent theory valid for some but not all cases) but they can originate also from ovarian parenchymal tissue.

The reviewer is correct: other epithelial ovarian cancers, like clear cell, endometrioid, mucinous, and others, have different origin. We only used samples from patients with high-grade serous cancer, which is the one that has been postulated to originate from the fallopian tube, mainly from the fimbria. The fimbria is location where we extracted RNA and DNA for the study. We clarified this in the manuscript and add a reference (#13).

  • Moreover, the normal tube is physiologically more involved than other tissues (i.e., ovary and endometrium) in infectious processes; then it seems physiological to finding a higher expression of BAVTs in normal fallopian tubes.

We agree with the reviewer that the tube has more apparent clinical impact after pelvic infections because its delicate balance for successful reproduction. However, we do not know the real impact of imbalances of the microbiome of the upper genital tract. There are few publications about changes in the microbiome and cancer. So, we do not fully understand the interaction of genital tract microbiome and disease. Gathering more and better information about this issue was one of goals of our research. We added additional information in the second paragraph of the discussion to emphasize this point (and ref #43).

  • Discussion: lines 356-358: how the authors can explain and support these two sentences with a likely opposite meaning.

We think that some of BAVT may be regulatory in nature, given the association with signaling pathways and some lncRNAs. Therefore, decrease in regulatory transcript may lead to increase in certain pathways that may facilitate growth and progression. We modified the text to make this point.

  • Lines 364-367: inflammation that characterizes the ovarian cancer microenvironment is a well-known factor involved in ovarian cancer carcinogenesis, independently from the changes in BAVTs. Indeed, inflammation is involved in ovarian cancer carcinogenesis mainly through the induction of DNA mutation mediated by the action of free radicals associated to inflammatory process. These mechanisms are well reviewed in Macciò et al Cytokine 2012.

We could not agree more with the reviewer. The point of these sentence was to stress the importance of inflammation in the process of carcinogenesis and the potential that an imbalance in the microbiome may be synergistic to the inflammation process created by endometriosis. We modified the text to clarify this point (and added Macciò reference, #46).

  • Lines 376-378: these sentence that introduce the more detailed list of main pathways below is not completely clear. In fact, although some pathways cited could be involved in an increased cancer risk if they are suppressed (i.e. IFN), others such as JAK and STAT-3 are involved in carcinogenesis and cancer progression and prognosis when they are hyperexpressed (similarly also EGFR and HGF). Therefore, it is not clear how lower expression of these BAVTs in cancers in comparison to normal tubes can be involved in carcinogenesis.

We deleted the sentence to avoid confusion. We only showed correlation with gene expression and lncRNAs, and association with pathways, no functional relation.

  • Some limitations, notably the risk of microbial contamination of samples, are very concerning and should be avoided by methodologically adequate procedures.

We do not believe that there was any contamination. All samples were processed under sterile conditions through a sequencing core that is regulated and guaranteed under the most stringent regulations. Furthermore, the results were validated in TCGA data, collected from some of the top research laboratories in the country. We do not think that they were contaminated either. However, we must consider all possibilities of biases in any research. We rewrote these sentences to clarify.

Reviewer 2 Report

The manuscript entitled “Bacterial, archaea and viral transcripts (BAVT) expression in gynecological cancers and correlation with regulatory regions of the genome” is well written and contain valid information.

But the manuscript lacks the essential details of the result, which might be exciting and scientifically meaningful.  

  1. The BAVT is associated with any regularity system or any biological pathways related to the pathology?
  2. What is the correlation/association of the IncRNA and how it can be used for the cancer therapy target?
  3. Competition of normal endometrium and benign endometrial with EEC might provide better concussion.  
  4. According to the author, the sample is obviously contaminated and must be avoided to get precise microbial analysis.
  5. The author should discuss the notable microbial population more clearly in the risk associated with cancer pathology and involvement in the progress of cancer

The manuscript needs more improvement in the analysis mythology and need major revision    

Author Response

REVIEWER 2

The manuscript entitled “Bacterial, archaea and viral transcripts (BAVT) expression in gynecological cancers and correlation with regulatory regions of the genome” is well written and contain valid information.

But the manuscript lacks the essential details of the result, which might be exciting and scientifically meaningful.  

  1. The BAVT is associated with any regularity system or any biological pathways related to the pathology?

We added more information about the PI3K-AKT/mTOR pathway as a regulator of tumor growth and progression in ovarian cancer (ref #53), and in endometrial cancer (ref# 55). Also added information about potential use of targeted therapeutics to this pathway in ovarian cancer (ref #54) and endometrial cancer (ref# 55). Additionally, we added information about EGFR TKI resistance (ref #58) and possible targeting treatment for its resistance (ref# 59).

  1. What is the correlation/association of the IncRNA and how it can be used for the cancer therapy target?

In the Discussion, we added more information about of the lncRNA, AL163541.1, and its association with some of the BAVTs and association with genes, like IFNA21, involved in pathways that lead to carcinogenesis and progression, like the PI3K-AKT/mTOR pathway. As we commented previously, we added information about potential targeted therapy to this pathway in ovarian and endometrial cancers (ref #54 and #55).

  1. Competition of normal endometrium and benign endometrial with EEC might provide better concussion.  

We intended to compare ovarian cancer and endometrial cancer. We used fallopian tube as control for ovarian cancer. For the present study, we had no availability of normal endometrium for RNA sequencing. Ideally, we would like to have a sample extracted and processed with the same conditions that the cancer samples to be comparable. We are asking for more funding to increase the number of cases and controls for both cancers.

We have clarified this concept throughout the manuscript and also made changes to represent the comparison between HGSC and ECC exclusively. Changes: Introduction, in the objectives description; Methods, section ‘Differences of BAVT expression between HGSC and EEC’; Results, sections ‘3.2. Differences of BAVT expression between HGSC and EEC’ and ‘3.3. Validation of BAVT analysis in TCGA dataset’; and Discussion..

  1. According to the author, the sample is obviously contaminated and must be avoided to get precise microbial analysis.

We apologize, the text was not clear. We do not think that there was any contamination. All protocols were followed with most strict analytical procedures. We have changed the paragraph to clarify this point.

  1. The author should discuss the notable microbial population more clearly in the risk associated with cancer pathology and involvement in the progress of cancer

We added a new second paragraph in the Discussion, detailing the evidence between our microorganism associated with HGSC and EEC and published associations with cell growth, some types of cancer and response to cancer treatment.

Reviewer 3 Report

The manuscript “Bacterial, archaea and viral transcripts (BAVT) expression in 2 gynecological cancers and correlation with regulatory regions 3 of the genome” by Bosquet et al. is interesting. However, I think they showed huge enthusiasm but wasted the potential. In my opinion, the title does not reflect the content. What do they mean when writing “Finally, BAVT may be associated with some gene and lncRNA regulatory systems involved in infectious and signaling pathways”? How does it refer to the title, which suggests some specific regulatory regions but not a general conclusion only? Similarly, when they summarize and conclude by the statement: “Some of these processes associated with BAVT could potentially be targets for cancer therapy” – what does it mean? What potential candidates do they suggest? What methods would be useful to use the data and perform the assessment? Altogether the manuscript seems too descriptive with moderate potential to indicate a therapeutic or diagnostic perspective? However, I find the analysis useful in the context of increasing knowledge on the contribution of ncRNA to carcinogenesis and metabolic dysfunctions.

Author Response

REVIEWER 3

The manuscript “Bacterial, archaea and viral transcripts (BAVT) expression in 2 gynecological cancers and correlation with regulatory regions 3 of the genome” by Bosquet et al. is interesting. However, I think they showed huge enthusiasm but wasted the potential.

In my opinion, the title does not reflect the content.

At request of other reviewers, we have added new elements in the Discussion (4th paragraph) to explain the association of BAVT and their correlated lncRNA with regulatory pathways associated with ovarian and endometrial cancer.

What do they mean when writing “Finally, BAVT may be associated with some gene and lncRNA regulatory systems involved in infectious and signaling pathways”?

We have modified the sentence to better reflect the results.

How does it refer to the title, which suggests some specific regulatory regions but not a general conclusion only?

We have better described those pathways that were associated with gene and lncRNA expressions correlated with BAVT. Some of these pathways are involved in cancer genesis, tumor progression, and response to cancer treatment. Obviously, these are associations which can point us to the right direction but will never demonstrate causality.

Similarly, when they summarize and conclude by the statement: “Some of these processes associated with BAVT could potentially be targets for cancer therapy” – what does it mean?

The statement was confusing. We modified it so is clearer that only functional analysis, no association analysis, would be able to identify proper targeted treatment.

What potential candidates do they suggest?

We added potential candidates of targeted therapies based on significant pathways associated to BAVT, throughout the 4th paragraph of the Discussion.

What methods would be useful to use the data and perform the assessment?

At the end of the 4th paragraph of the Discussion we described how we would think it could be studied. First with functional analysis at the protein level on how changes in BAVT expressions affect those pathways associated with them. Then to assess which are the areas that could be targeted with novel therapies, based on these functional analyses.

Altogether the manuscript seems too descriptive with moderate potential to indicate a therapeutic or diagnostic perspective?

We added some therapeutic perspectives in the Discussion, as previously described.

Round 2

Reviewer 1 Report

The authors improved the manuscript addressing several reviewers' comments. However the methodological concerns remain. 

Reviewer 2 Report

The manuscript is improved well, and it can be published. 

Reviewer 3 Report

The manuscript was significantly improved.